

# Quality of experience-aware application deployment in fog computing environments using machine learning

P. Jenifer[1] and J. Angela Jennifa Sujana[2]

[1] Computer Science and Engineering, Francis Xavier Engineering College, Tirunelveli, Tamil Nadu, India

[2] Artificial Intelligence and Data Science, Mepco Schlenk Engineering College, Sivakasi, Tamil Nadu, India

Corresponding author
P. Jenifer,
jeniferjebavaram@gmail.com

## ABSTRACT

Edge intelligence is fast becoming indispensable as billions of sensors demand real-time inference without saturating backbone links or exposing sensitive data in remote data centres and emerging artificial intelligence (AI)-edge boards such as NVIDIA CPUs, 16 GB RAM, and microcontrollers with chip neural processing unit (NPU) (<1 W). This article introduces the Energy-Smart Component Placement (ESCP) algorithm of fog devices like fog cluster manager nodes (FCMNs) and fog nodes (FNs), allocates modules to fog devices, and saves energy by deactivating inactive devices framework transparently distributes compressed neural workloads across serverless. To optimize the deployment of AI workloads on fog edge devices as a service (FEdaaS), this project aims to provide a reliable and dynamic architecture that guarantees quality of service (QoS) and quality of experience (QoE). The cloud, fog, and extreme edge layers while upholding application-level QoS and QoE. Two machine learning (ML) methods that fuse eXtreme Gradient Boosting (XGB)-based instantaneous QoS scoring and long short term memory (LSTM) forecasting of node congestion, and a meta-heuristic scheduler that uses XGB for instantaneous QoS scoring and LSTM for short-horizon load forecasting. Compared with a cloud-only baseline, ESCP improved bandwidth utilization by 5.2%, scalability (requests per second) by 3.2%, energy consumption by 3.8% and response time by 2.1% while maintaining prediction accuracy within +0.4%. The results confirm that low-resource AI-edge devices, when orchestrated through our adaptive framework, can meet QoE targets such as 250 ms latency and 24 h of battery life. Future work will explore federated on-device learning to enhance data privacy, extend the scheduler to neuromorphic processors, and validate the architecture in real-time intensive care and smart city deployments.

## INTRODUCTION

Due to the widespread use of cutting-edge technology and the Internet of Things (IoT), several advanced gadgets are interconnected inside edge-assisted IoT networks to meet the

diverse technical requirements of modern living (*Balasubramanian et al., 2019*). Moreover, AI facilitates the enhancement of user-friendliness in these technologies. Internet of Things (IoT) technologies using edge computing get advantages from AI since it automates and optimizes their functionalities (*Lu et al., 2020*). The integration of machine learning (ML) with edge computing has resulted in a novel concept: "edge intelligence" (*Deng et al., 2020*; *Hayyolalam et al., 2021*). Innovative technologies, including autonomous vehicle systems, smart device surveillance, real-time critical infrastructure management, and predictive healthcare, are poised to develop from this concept (*Hayyolalam et al., 2021*).

Cloud computing may be unsuitable for real-time applications like online gaming and video streaming due to data source distance (*Abdullah & Jabir, 2021*). Object-level devices lack processing power and storage for many tasks. Constraints prevent launching fully developed apps. Cloud applications and algorithms may need plenty of processing power. This technique has security hazards connected to distant processing and data transmission, connectivity issues to remote cloud locations, and delays between IoT devices and cloud nodes. Compact data centres with minimal latency store operational data on fog devices. IoT devices may integrate algorithms and apps when connected to cloud nodes. Fog computing and storage provide high-bandwidth, low-latency services (*Ghaleb & Farag, 2021*; *Confais, Lebre & Parrein, 2020*; *Karagiannis & Schulte, 2020*). Cloud computing experts are improving frameworks to address these issues. Cloud data processing costs must be considered. The service may save energy and time, but high pricing may prevent customers (*Zhu et al., 2017*). Fog computing allows devices to do complex operations at the network edge, making it useful for IoT mobile application processing. Fog devices need complex scheduling and resource allocation to assign jobs (*Jiang et al., 2019*). Cloud task offloading continues because fog computing has fewer resources than cloud computing. Identifying and allocating IoT device work offloading requires optimal solutions in a hybrid IoT, fog, and cloud environment. We provide energy-efficient intelligent job offloading for IoT, fog, and cloud computing. This multi-classifier system calculates the task, network, and processing properties of fog nodes (FNs) to choose the best service node. Offloading request situations are improved by the multi-classifier system by considering energy utilisation, data transfer speed, service request transmission, and execution time.

We want to give an effective and environmentally friendly way to offload work to the cloud, fog, and IoT. The multi-classifier system will excel in cloud, fog, IoT processing node, network, and task characteristics. These characteristics will classify and choose work nodes based on service cost, energy usage, and transfer and execution time. The following are the main contributions of this article:

- To use multi-classifier ML, propose and create a new model for job offloading choices in an Energy-Smart Component Placement (ESCP) algorithm for fog-cloud computing architecture.
- To analyse and assess the suggested model by contrasting it with previous efforts; moreover, to show how the suggested method uses machine learning (ML) techniques to

improve quality of service (QoS) and quality of experience (QoE) metrics, namely application time to response, network utilisation, and consumption of energy.

- To address the integration problem of heterogeneity by ensuring that application processing and node interaction are platform-independent.
- To provide a system model for assessing the effectiveness of cloud, fog, edge, and serverless computing in economic and medical apps using the IoT, employing advanced ML models for long short term memory (LSTM) and eXtreme Gradient Boosting (XGB).

## RELATED WORKS

The IoT includes hardware, software, sensors, and components for computers, among other things, that facilitate the collection and sharing of data for processing by other IoT devices and systems (*Gill et al., 2019*). IoT devices include, but are not limited to, fitness trackers, smartwatches, and medical sensors. The goal of this subfield of AI, as stated, is to streamline decision-making processes by eliminating the need for human analysts by automating data analysis techniques for trend prediction. This Internet-based on-demand service allows several cloud customers to access data and compute-intensive resources *via* a specified user interface. Software, platforms, and infrastructure are the three main categories of services offered by cloud computing (*Aslanpour et al., 2021*). This decentralised strategy tries to bridge the gap between the IoT devices and the cloud data centre. Its goal is to enhance the cloud's capabilities and speed up reaction times for time-sensitive IoT applications. This model of distributed computing reduces latency and increases reaction time by bringing data storage and computational resources nearer to edge/IoT devices (*Singh, Singh & Gill, 2021*).

An important part of resource management in fog computing is load balancing, which ensures that all processes and applications are given an equitable share of the available resources. When applied to fog computing environments, conventional load-balancing methods fail because of their inflexibility and lack of uniformity (*Omoniwa et al., 2018*). Service quality guarantees in fog computing systems are, therefore, very challenging. There has been a plethora of studies on fog computing during the last two decades. To make better use of the fog environment, one might study certain scholarly articles. Improving the area of QoS aspects is possible. While providing helpful background on several models and algorithms, the article delves more into the approaches and strategies recommended for enhancing adaptation in fog circumstances (*Brogi & Forti, 2017*; *Ni et al., 2017*). A decentralised approach to service design using blockchain technology has been proposed by academics in *Al Ridhawi et al. (2020)* for the domain of offering complex multimedia services to clients in the cloud. The suggested approach does away with transitional services and network provider units by dynamically creating user-defined authentication and composite service provisioning services. This study presents a trustworthy, scalable, adaptable, and decentralised cloud infrastructure. By combining state-of-the-art services with technologies like software-defined networking (SDN), blockchain, and fog computing, it offers a sophisticated and cutting-edge solution right at the router (*Al Ridhawi et al., 2018*). A separate research project aimed at developing an effective

composite service delivery system by introducing a framework for context-aware real-time cooperation at the network edge. Devices used by end users and mobile edge clouds (MECs) will be part of this design. Duplicating data and services hosted in the cloud over several MEC nodes is the suggested approach. The authors built a privacy-focused AI task generation framework to simplify edge AI deployment. They propose task offloading and distribution to accomplish complex AI tasks on edge devices. The Skyline optimisation approach provides an intelligent service selection mechanism (*Rahman et al., 2020*).

The work presents a QoS-aware cloud-edge service discovery and selection model in an IoT setting, with an emphasis on the outcomes indicated above. To evaluate QoS factors as non-functional attributes, this model employs a hybrid multi-objective metaheuristic method based on a grey wolf optimiser and a genetic algorithm (GWO-GA). The proposed approach for the service discovery and selection issue in the IoT context seeks to ensure QoS requirements such as response time, energy use, and cost concerns. According to trials, the proposed technique surpasses the other algorithms by 30% when it comes to cost reduction (*Wang & Lu, 2021*). This approach prioritises both energy efficiency and security. The proposed method makes use of a GA and a multi-objective GWO. Energy efficiency, response speed, and the costs of discovering and selecting services in the IoT are all much enhanced by the proposed hybrid approach. The authors of *Huang, Liang & Ali (2020)* have also presented an optimisation method for service composition that uses simulations to improve reliability. They have taken a two-layer approach to the system analysis, looking at both the edge and cloud levels. They created a model aggregation strategy for problems with service composition and used a probabilistic Petri net model to build both layers. To provide an amalgamated service under typical circumstances, most conventional methods of rearranging service composition centre on service scheduling; nevertheless, these methods are unable to quickly adjust to changing environmental conditions. Therefore, to address the stated problem, the authors of *Gao, Huang & Duan (2021)* proposed a dynamic reconfiguration of service processes in mobile edge e-commerce environments. Despite the evident advantages of fog computing for medical treatment monitoring systems and numerous studies validating the foundational concepts of this computing paradigm, formulating strategies to optimise the utilisation of fog devices remains challenging due to their constrained power and computational capacities, as well as the need to appropriately allocate modules to specific fog devices to attain the intended level of service (*Hassan et al., 2022*). Our thorough performance analysis was made possible by the iFogSim simulator, which uses the Energy-Efficient Internet of Medical Things (EEIoMT) to Fog Interoperability of Task Scheduling architecture (*Alatoun et al., 2022*). Using the proposed method, our study assesses the information distribution, resource management, and job scheduling of the Fog Computing Layer.

## METHODOLOGY

For cloud-based systems, it is standard procedure to overprovision resources to prevent a decline in customer service in the event of an unforeseen calamity. This results in excessive power utilization as well as wasteful expenditure. A cloud-hosted system is made up of RAM, storage space, processor power, and I/O functions. Even while certain optimization

techniques lower resource use, they may have a detrimental effect on service quality (such as accessibility or response time). It is consequently not easy to determine the ideal quantity of cloud resources in terms of both cost and QoS. We are interested in researching systems that use the most popular cloud service models, such as Infrastructure as a Service (IaaS), Platform as a Service (PaaS), and FEDaaS, to deliver applications. Both the amount and quality of cloud components may be scaled individually by these models without affecting each other. Additionally, they can optimize themselves since they are dispersed throughout several cloud regions. To keep things simple, we will only demonstrate our approach in a single application. This integrated system concept combines several software and hardware elements to provide platform independence and organized communication.

## Cloud datacenter

IoT back-end apps are operated in the cloud when the capacity with fog architecture is not enough for handling a program or when applications that require latency are operating. With this method, the system model may be used to investigate the computing resources for IoT applications. Serverless technology is used in it. Compute, storage, virtual machine (VM) management, and resource scheduling are some of the most important components of a cloud data centre. While the latter two are in charge of managing virtual computers, the former two are in charge of scheduling both real and virtual resources. Cloud customers can build and launch IoT apps and services without worrying about server administration on the serverless platform, which links cloud data centres with fog infrastructure. It provides adaptable scalability and affordable deployment of IoT applications. The four main parts of the Serverless platform are provisioning, computing, monitoring, and storage. Provisioning is responsible for allocating resources to meet user requests, computing is responsible for doing computations, and storage is responsible for retaining data for processing. The serverless platform consists of a data manager, a resource manager, a ML model, and a security administrator. Data collected from various IoT devices is overseen by the data manager before it is processed further. To complete a job, the resource manager must first gather all of the required materials and then distribute them. Data is used to build ML models that can anticipate trends that meet the needs of an IoT application. To provide adequate protection, the security manager uses a variety of security techniques.

## Fog architecture

The formation of the fog architecture is the result of two essential elements. These consist of the fog computational centers and fog gateway nodes (FGNs).

### Fog gateway nodes

The FGNs function as the portal to the realm of distributed computing. The proposed framework indicates that the FGNs assist IoT devices with employment placement, along with application processing. The FGN provides an interface for supplementary applications, including controlled IoT devices, credential authentication, backend program access, resource requirements for the processing of applications, and service expectation articulation. Furthermore, FGN formats the data uniformly and sanitizes it. After data

collection from many sources, aggregate data is collected. For substantial processing, the data is transferred to supplementary computer nodes *via* a combined environment in Eq. (1).

$$P_{FGN} = \lambda \cdot R_{\text{comp}} + \mu \cdot R_{\text{comm}} + v \cdot E_{\text{latency}} \tag{1}$$

where, $P_{FGN}$ is the performance metric, $R_{\text{comp}}$ is the computational resource consumption (*e.g.*, central processing unit (CPU)/graphics processing unit (GPU) usage) rate, $R_{\text{comm}}$ is communication efficiency (*e.g.*, bandwidth, packet transfer), $E_{\text{latency}}$ is an energy-delay factor (*e.g.*, time delay in ms), and $\lambda$, are weighting coefficient.

In Eq. (1), the variable is defined as follows:

$\lambda, \mu, v \in [0, 1]$ are normalized weighting factors used to balance the influence of computation, communication, and latency, respectively. The sum can be considered to be 1 for normalization (*i.e.*, $\lambda + \mu + v = 1$).

$R_{\text{comp}} \in [0, R_{\text{max}}]$, where $R_{\text{max}}$ is the maximum available computation resource in the FGN, typically measured in FLOPS or CPU units.

$R_{\text{comm}} \in [0, B_{\text{max}}]$, where $B_{\text{max}}$ is the maximum communication bandwidth or throughput available.

$E_{\text{latency}} \in [0, L_{\text{max}}]$, where $L_{\text{max}}$ is the maximum tolerable or measured latency, typically in milliseconds or seconds.

### Fog computational nodes

The proposed design was developed to manage the massive number of FCNs running in parallel. It is the job of FCNs to design new storage capacity and resource designs. Processing cores, memory, storage, and bandwidth are the building blocks of FCNs, which are used to carry out processing tasks. Here are the responsibilities that FCNs perform in Eq. (2).

$$P_{FCN} = \alpha \cdot U_{cpu} + \beta \cdot B_{\text{comm}} + \gamma \cdot E_{\text{energy}} \tag{2}$$

where, $P_{FCN}$ represents node performance, $U_{cpu}$ is CPU utilization, $B_{\text{comm}}$ is bandwidth usage, $E_{\text{energy}}$ is energy consumption, and $\alpha$, $\gamma$ are weights for each factor.

In Eq. (2), the variable is defined as follows,

$\alpha, \beta, \gamma \in [0, 1]$; these are normalized weighting factors assigned to CPU usage, communication bandwidth, and energy consumption, respectively. The values can be set according to the optimization priority and may satisfy $\alpha + \beta + \gamma = 1$ for normalization.

$U_{cpu} \in [0, 1]$, CPU utilization ratio of the fog computing node, expressed as a fraction or percentage (*e.g.*, 0.75 = 75% CPU usage).

$B_{\text{comm}} \in [0, B_{\text{max}}]$, Bandwidth consumption or data transmission requirements, typically measured in Mbps or similar units. $B_{\text{max}}$ is the maximum available bandwidth.

$E_{\text{energy}} \in [0, E_{\text{max}}]$, Energy consumed by the fog computing node, usually measured in joules (J) or watts (W), depending on the context. $E_{\text{max}}$ is he an upper bound on acceptable or expected energy consumption?

Figure 1 illustrates a multi-layered model designed to enable intelligent application deployment in fog computing environments, particularly for IoT-based use cases. At the

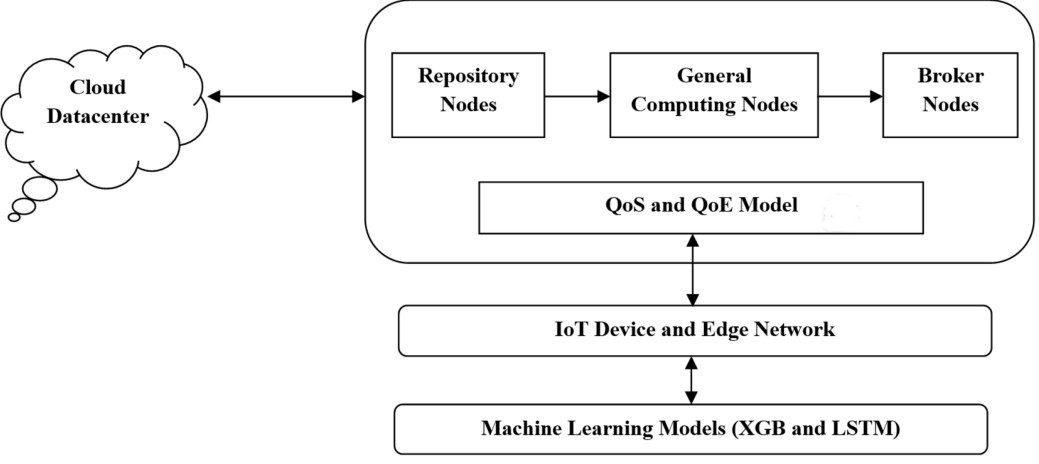

**Figure 1 Proposed system architecture.**

base layer, ML models like XGB and LSTM handle tasks such as performance prediction, workload forecasting, and decision-making. These models directly interface with IoT devices and the edge network layer, which includes a variety of low-power sensors, wearables, medical monitors, and routers responsible for real-time data generation and initial edge processing.

The Fog architecture layers serve as an intermediate processing environment between the edge device and the cloud. It includes:

- QoS and QoE models, which access network and user-level performance,
- Broker nodes (BNs), which handle task scheduling and service matching.
- General computing nodes (GCNs), which execute moderately intensive tasks closer to the data source,
- Repository nodes (RNs), which manage data storage and resource availability.

At the top, the cloud data center layer is responsible for handling large-scale data analytics and long-term storage when local resources are insufficient.

This proposed architecture ensures reduced latency, improved energy efficiency, and enhanced responsiveness by enabling smart task offloading across layers. It supports scalability and reliability for critical IoT applications, especially in dynamic environments.

The RNs oversee the replication, data sharing, remuneration, and storage security functions of the distributed database. Registered nurses provide access to current data and the study of past data. Generated and maintained application-specific metadata, including dependencies, models, and processing needs. Nevertheless, the data culminates at these nodes, since they are the source of run-time data for anomaly-driven applications.

Not all FCNs have instant access to FGNs. The FCN and the FGN are connected *via* the BN. These BNs are responsible for managing resources and submitting applications for processing, along with the required information. Multiple BNs may be serviced simultaneously by GCNs with reliable performance. When distributed applications are

running, an automatic group of GCNs is created underneath the BN. Available FCNs with FGN collaboration allow the back-end functioning of IoT applications. After FCNs have allocated enough resources, they begin running the application's backend. If the FGN is unable to run the back-end application due to a lack of resources, the FCN will step in and provide them, much like a BN. Other FCNs and cloud data centres are its interfaces. This leads to delegating tasks to different FCNs and then arranging, overseeing, and coordinating their activities. To back up these BNs, the suggested architecture makes use of deep learning to spot anomalies, blockchain to add security measures, and replication to make it fault-tolerant. Secure communication between cloud data centres, FGNs, and FCNs is made possible by this resilient architecture.

## Quality of service and quality of experience model

For every edge service (device), this article examines four QoS metrics: availability, reliability, latency, and reaction time. The values of the variables we utilised were sourced from the Quality of Web Service (QWS) dataset, which is a popular resource for academic research on service composition difficulties. Out of the six QoS measures that were utilised to evaluate the 2,507 valid services in this study, four are used in this report. The baseline values of the QoS parameters are determined using the dataset. With each service invocation, the QoS values are randomly changed to mimic the non-static nature of QoS values in real-world circumstances and to promote dynamic service composition. With every call to the edge service, the QoS and QoE metric values fluctuate wildly; to keep track of them, we included a monitoring component inside the framework. At first, we take the number of subjective evaluations (K) and use them to construct K randomly assigned QoE values for all the services in the dataset. This means that for every service, K people provide measurable feedback. With every service call, the QoS and QoE parameters for the relevant service would be changed at random. Because there is a linear relationship between QoE and QoS values, and because these QoS values may vary, the new QoS parameters will be used to modify the QoE value. We include the computed QoE values in the dataset as an additional QoS measure, and the technique is completely stochastic. It generates a number at random from 0 to 1.

$$QoE = \alpha \cdot QoS_{\text{latency}} + \beta \cdot QoS_{\text{throughput}} + \gamma \cdot QoS_{\text{packet loss}} + \delta. \tag{3}$$

In Eq. (3), the variable is defined as follows:

$\alpha, \beta, \gamma \in [0, 1]$; these are the weighting coefficients that determine the relative importance of latency, throughput, and packet loss in contributing to overall QoE. Often, they satisfy $\alpha + \beta + \gamma = 1$ for normalization.

$QoS_{\text{latency}} \in [0, 1]$, A normalized value representing latency-based QoS. Lower latency (closer to 0) improves QoE, so the normalized form could be:

$$QoS_{\text{latency}} = 1 - \frac{latency}{L_{max}}, \text{ where latency} \leq L_{max}.$$

$QoS_{\text{throughput}} \in [0, 1]$, A normalized throughput score. Higher throughput improves QoE, so: $QoS_{\text{throughput}} = \frac{throughput}{T_{max}}$, where throughput $\leq T_{max}$.

$QoS_{\text{packet loss}} \in [0, 1]$, Reflects packet loss quality, since lower packet loss is better, it is normalized as: $QoS_{\text{packet loss}} = 1 - \dfrac{\text{packet loss}}{P_{max}}$, where packet loss rate $\leq P_{max}$

$\delta \in \mathbb{R}$, A bias or adjustment term that allows for baseline scaling or offset in the final QoE-score, often used to fine-tune the output.

## IoT device and edge network

In fog computing environments, IoT devices form the primary data generators and play a crucial role in influencing application QoE. These devices include a wide range of sensors, actuators, smart cameras, wearables, and embedded systems that continuously produce real-time data across domains such as healthcare, smart homes, and industrial automation. However, these devices are resource-constrained, having limited computation, memory, and energy capacity. To support intelligent QoE-aware application deployment, edge networks act as intermediaries between IoT devices and fog/cloud infrastructure. Edge nodes, such as local gateways, micro data centres, and edge servers, perform lightweight processing tasks such as initial data filtering, aggregation, and inference using pre-trained ML models. These devices reduce latency, bandwidth, and consumption and centralise load, thereby improving user-perceived QoE. In the proposed system, the edge layer supports the deployment of ML algorithms (*e.g.*, QoS prediction, resource estimation) and collaborates with FGNs to make localised decisions based on real-time conditions. The communication protocols between IoT devices and the edge layer leverage low-power and high-speed technologies such as 5G, Wi-Fi 6, NB-IoT, and LoRaWAN. Ensures low-latency, energy-efficient, and scalable data handling, which is essential for dynamic, QoE-driven application deployment in fog environments.

## Machine learning models

This section takes the preprocessed data from IoT devices and utilises it to train artificial intelligence models to classify data points. The creation of these feature vectors follows the data-collecting process. Prior work made use of the graphical user interface (GUI) to make ensemble voting-based data state predictions. With this project in mind, we used state-of-the-art ML methods like XGB and LSTM. The workload manager is in charge of processing data, managing incoming job requests, and allocating tasks in a queue. The arbitration module arranges fog/cloud resources for each task's execution based on its QoS requirements. Depending on the demands of the user, the broker decides whether to execute a job at a cloud node or a FN. This system has a credential archive where users' credentials for authentication are kept. Additionally, the credential archive gives the security keys to other parties and alerts them when the broker service creates a new data block.

In the proposed QoE-aware application deployment framework, two advanced ML models, LSTM and XGB, are strategically integrated to make intelligent, adaptive deployment decisions in a fog computing environment. This work is designed to enhance QoE by proactively forecasting resource usage trends and evaluating current system conditions. The LSTM model is employed to capture temporal dependencies in system

performance metrics. It is trained on historical time-series data collected from IoT, edge, fog, and cloud nodes, including CPU utilisation, memory usage, latency, queue length, and bandwidth status. The input to the LSTM is a sequence of N past metrics per node, and the output is a future horizon of predicted system states (*e.g.*, expected CPU load or delay for the next 5–10 time intervals). This temporal foresight allows the system to anticipate bottlenecks or overloads before they happen, enabling preemptive job offloading to alternative nodes with better predicted performance. The LSTM consists of two stacked layers with 64 memory units each, followed by a dense layer that outputs a prediction vector. During training, the model is optimised using the ADAM optimiser and validated using mean absolute error (MAE) to ensure robust short-term forecasting. Complementing the LSTM, the XGB model functions as a high-performance regressor or classifier that processes current, real-time features to score or rank the suitability of each node for task deployment. The features fed into XGB include instantaneous values such as network latency, available bandwidth, packet loss, and node energy consumption, along with contextual metadata such as application priority, task size, and time-of-day indicators. XGB ensemble of decision trees is turned using a hyperparameter search over tree depth, learning rate, and subsample ratios. The output is a score that reflects the node's current fitness for hosting an application, normalised between 0 and 1. Together, the LSTM and XGB models are fused in a multi-criteria decision engine. For every candidate deployment node, the system aggregates (i) the LSTM's predicted performance horizon, (ii) the XGB's real-time suitability score, and (iii) the current resource availability (*e.g.*, CPU, memory, and bandwidth). These inputs are combined in a weighted utility function. The orchestrator then selects the node with the highest utility score for application deployment. If no nodes meet a minimum utility threshold (including potential QoE degradation), a rule-based fallback selects the nearest edge node or defers the task to a cloud node.

The intelligent offloading mechanism ensures that applications are always placed in the most appropriate execution environment, whether edge, fog, cloud, or serverless, while optimising for QoE metrics like response time, throughput, network utilisation, and energy consumption. Furthermore, the platform-agnostic design of the ML models (served *via* Open Neural Network Exchange (ONNX) or container-based microservices) ensures they can be deployed across heterogeneous hardware and software environments, addressing one of the key challenges in fog computing: interoperability and heterogeneity. Table 1 describes the summary of ML model roles.

### eXtreme Gradient Boosting

XGB, is an effective ML approach for evaluating QoE and QoS in cloud service settings. Its capacity to manage extensive datasets, intricate relationships, and absent values makes it optimal for pattern identification and performance metric forecasting. XGB employs gradient boosting to generate exceptionally precise models while maximising resource allocation, minimising latency, and enhancing user happiness. The extensive scale facilitates real-time analysis, allowing cloud providers to perpetually enhance their

| Table 1 | Summary of ML model roles. | | |
|---------|---------|---------|---------|
| **Model** | **Role** | **Input** | **Output** |
| LSTM | Predict future node performance | Time-series metrics (last N timesteps) | Predicted load/delay |
| XGBoost | Rank the current node's suitability | Instant QoS metrics + context | QoE-aware deployment score (0–1) |
| Fusion layer | Combine LSTM, XGB, and resource metrics | Forecast + score + real-time state | Offloading decision (edge/fog/cloud/serverless) |

---

**Algorithm 1 Algorithm for XGB.**

---

**Input:** Feature matrix F (instant QoS + context), label/target T

      where T is either (a) a regression QoS score or (b) an offload class

**Output:** Trained model M_XGB accessible *via* Score()

**Step 1:** Pre-process:

      a. One-hot encode categorical context features

      b. Scale/normalize continuous QoS inputs

**Step 2:** Hyperparameter search (*e.g.*, grid or Bayesian):

      For each candidate config C:

          train XGBoost with parameters C (max_depth, n_trees, η, subsample)

          evaluate metric (RMSE for regression; F1 for classification)

      Select config C* with the best validation metric

**Step 3:** Retrain the full model on the combined Train+Validation set with C*

**Step 4:** Persist trained model as M_XGB

**Step 5:** Deploy:

      Wrap M_XGB in a stateless microservice

      Expose Score(request_features) → scalar S_QoS or class label

---

services. Integrating XGB with QoE and QoS frameworks enhances decision-making, dependability, and performance in highly competitive cloud computing settings.

Algorithm 1 describes the XGB QoS model is fitted to instantaneous features. Categorical context (such as workload type and hour of day) is one-hot encoded, while numeric QoS readings (latency, throughput, energy, and packet loss) are normalised. A systematic hyperparameter search chooses the optimum tree depth, learning rate, and ensemble size by maximising validation F1 (for classification) or minimising Root mean square error (RMSE) (for regression). The selected configuration is retrained on the full training split, saved, and exposed as a stateless microservice that converts a fresh QoS features vector into either a normalised suitability score or an offloading class label.

### Long short term memory (LSTM)

It is reasonable to use LSTM, recurrent neural networks (RNNs) for cloud QoE and QoS prediction since these networks excel at simulating sequential data. Service effectiveness and customer satisfaction patterns may be correctly predicted by LSTMs due to their proficiency in recognizing long-term dependencies. They could look at time-series data like error rates, capacity utilization, and network latency to foretell potential abnormalities or bottlenecks. Incorporating LSTM models may help cloud service providers optimize

| Algorithm 2 Algorithm for LSTM. |
|---|

**Input:** Historical metric logs D, horizon H, sequence length N

**Output:** Trained model M_LSTM accessible *via* Predict()

**Step 1:** Pre-process:

    a. Remove missing records, interpolate short gaps

    b. Scale each continuous feature to [0, 1]

    c. Segment D into overlapping sequences of length N → X and their next-H targets → Y

**Step 2:** Initialize LSTM network:

    Layer1: LSTM(units=64, return_sequences=True)

    Layer2: LSTM(units=64)

    Dense(H × |features|) with linear activation

**Step 3:** Split X, Y into Train/Validation/Test (80/10/10)

**Step 4:** Train using Adam optimizer:

    Repeat until early-stopping

    compute loss = MAE(Y_pred, Y_true)

    Back-propagate loss, update weights

**Step 5:** Evaluate on Test; persist best weights as M_LSTM

**Step 6:** Deploy:

    Wrap M_LSTM in a lightweight inference service

    Accept REST/IPC request ⟨node_id, last N snapshots⟩

    Return the forecast vector of size H

resources, enhance the user experience, and maintain consistent service quality, all of which may lead to increased customer retention and productivity in operation.

Algorithm 2 describes the LSTM forecaster is prepared by cleaning the historical telemetry logs, normalizing every continuous metric to a 0–1 scale, and sliding a window of N consecutive shots to build input-target pairs. A two-layer LSTM with 64 hidden layers learns to map each sequence to a horizon of H predicted values (future CPU load, queue depth, and link delay). Training uses the ADAM optimizer until early stopping on the validation set minimizes MAE; the best weights are exported and wrapped in a lightweight inference service that returns a forecast vector of the latest N readings for any nodes. During real-time offloading, the orchestrator queries every candidate node in parallel. It feeds each node's most recent N telemetry shots into the LSTM service to obtain short-term forecasts and, at the same time, passes the current QoS vector to the XGB service to obtain an instantaneous suitability score. These two predictions are combined with the live resource vector (CPU, memory, bandwidth) in a weighted utility function. Nodes that cannot satisfy hard constraints, such as minimum free memory, are discarded, and the destination with the highest utility is selected.

$$P_{FGN/FCN} = \lambda \cdot XGB(P_{QoS}) + \mu \cdot LSTM(P_{time-series}) + v \cdot R_{resource} \tag{4}$$

where, $P_{FGN/FCN}$ is the node performance, $XGB(P_{QoS})$ models QoS metrics using

XGB, $LSTM(P_{\text{time}-\text{series}})$ predicts time-dependent trends using LSTM, $R_{\text{resource}}$ is resource utilization, and $\lambda, \mu, \nu$ are weight coefficients.

In Eq. (4), the variable is defined as follows:

$\lambda, \mu, \nu \in [0, 1]$; these are the weighting factors controlling the influence of QoS prediction, time-series performance trends, and resource usage, respectively. They may be normalized such that, often, they satisfy $\lambda + \mu + \nu = 1$ for normalization.

$XGB(P_{QoS}) \in \mathbb{R}$, this term represents the output of an XGB model trained on QoS metrics (latency, throughput, jitter, *etc.*). The value is typically normalized or scaled based on the prediction (*e.g.*, QoS score between 0 and 1 or a regression score for priority).

$LSTM(P_{\text{time}-\text{series}}) \in \mathbb{R}$ The LSTM model outputs, which forecast system performance (*e.g.*, load, delay, failure probability) over time. Its output is based on historical performance trends, typically in the normalized range [0, 1] or in units relevant to performance (*e.g.*, expected delay in ms).

$R_{\text{resource}} \in [0, R_{\text{max}}]$, resource consumption (CPU, memory, bandwidth, *etc.*) in the FGN/FCN. This can be scalar, representing either a composite resource score or a key individual metric like CPU utilization. It is often normalized between 0 and 1 or bounded by a system-defined upper limit.$R_{\text{max}}$.

## RESULTS AND DISCUSSION

The suggested AI subtask composition approach was evaluated using Python on a laptop with an Intel Core i7-10750H CPU running at 2.60 GHz and 16 GB of random access memory (RAM). To provide a fair comparison, all three techniques were fed the identical set of services collected from the dataset. Every trial's QoS values were derived using the QWS dataset. During the composition process, the QoS values are randomly updated to offer a dynamic composition framework. Four qualities of service metrics, availability, dependability, response time, and latency, have been the primary focus of our data-collecting efforts.

### Quality of Web Service (QWS) dataset

The first dataset of its sort to measure the QoS of real internet services was created in 2007 as part of Eyhab Al-Masri's PhD thesis. Dataset available at open source download link https://qwsdata.github.io/. In 2007, it was introduced. The QWS Dataset has been downloaded over 9,000 times since it was first made available in 2007, demonstrating how well accepted it is among academics. This dataset might serve as a starting point for researchers who are interested in online services. Search engines, service portals, Universal Description, Discovery, and Integration (UDDI) registries, and other publicly accessible online resources were the primary sources of the web services that were gathered using the Web Service Crawler Engine (WSCE). In 2008, tested the QWS of 2,507 web services using our web service broker (WSB) technology for the QWS Dataset ver. 2.0. Each row of this collection contains nine QWS metrics for each web service, separated by commas. The first nine components, which are QWS metrics, were evaluated over 6 days using a variety of Web service benchmarking tools. The average readings for all the measurements made

during that period are the QWS values. The last two inputs are the service name and the Uniform Resource Locator (URL) of the Web Services Description Language (WSDL) file.

## Energy-conscious methods

Our energy-efficient strategy aims to reduce the overall energy consumption of all fog devices by implementing an optimised resource allocation and job scheduling system for IoT devices inside a fog ecosystem. Our methodology integrates Fog Cluster Manager Nodes (FCMNs) and FNs with dynamic classification using the suggested strategy, facilitating effective resource allocation and optimising response time. Unlike those that focus on energy consumption in scheduling tasks, our methodology employs distributed protocols for efficient coordination and communication, therefore substantially reducing energy consumption and improving sustainability in fog-based healthcare settings. Our system dynamically locates the nearest gateway to decrease latency, unlike earlier research that allocated end devices to gateways or used cloud-based data processing. The ESCP algorithm and clusters of fog devices like FCMNs and FNs helped us allocate modules to fog devices and save energy by deactivating inactive devices. This section examines our trial evaluation criteria and how effectively our method tackles latency and energy utilization issues.

## Energy-smart component placement (ESCP)

In the ESCP Algorithm, the Data Processing Module is positioned strategically among the FNs that are linked to every Fog Cluster Management Node (FCMN). The Data Processing Module is strategically arranged for optimal system performance and energy efficiency with the aid of a thorough assessment of FN computing capabilities and energy levels. Each FCMN classifies FNs into distinct energy levels based on a predefined threshold, allowing for a detailed analysis of each node's processing power. This evaluation is used by the ESCP algorithm to identify the Data Computing Module's position, ensuring that the selected FNs have the energy and computing power needed to carry out IoT device operations efficiently.

An essential performance statistic is the average end-to-end latency, which shows the time it takes for operations to go from the device itself to the fog layer and back again. Keeping the intended QoS standards in place is critical and has an immediate impact on the system's response to critical patient data. The total end-to-end delay contains the latency elements processing time (PT), transmission time (TT), and queue time (QT). This delay encompasses the whole system's sequence of activities.

Time spent processing tasks by FNs is denoted by PT, data transfer between end devices is denoted by TT, and queueing time is denoted by QT. Having a complete grasp of these components makes it simpler to identify opportunities for latency improvement in the proposed architecture.

We compared our proposed model's average latency to that of cloud-based and latency-aware solutions. The measurement of delay is in milliseconds. By taking advantage of the closeness of fog devices to end devices to reduce end-to-end latency, our approach outperforms cloud-based and latency-aware models. Both our proposed model and the

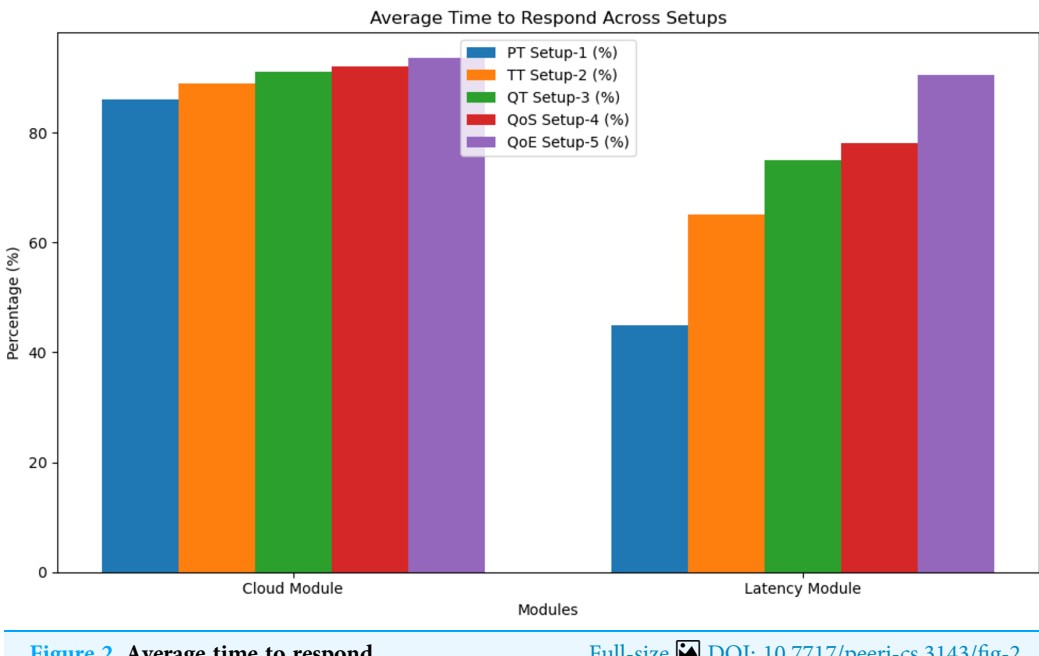

**Figure 2  Average time to respond.**

**Table 2  Average time to respond.**

| Module | PT Network Setup-1 (%) | TT Network Setup-2 (%) | QT Network Setup-3 (%) | QoS Network Setup-4 (%) | QoE Network Setup-5 (%) |
|---|---|---|---|---|---|
| Cloud module | 86 | 89 | 91 | 92 | 93.5 |
| Latency module | 45 | 65 | 75 | 78 | 90.5 |

latency-aware model use fog devices close to end devices to perform a task, which considerably reduces average end-to-end latency as compared to the cloud-based technique. In Fig. 2, we can see that there have been continuous improvements in the average latency of our suggested model, the cloud-based model, and the latency-aware model. Table 2 shows the percentage reduction over cloud-based and latency-aware models across different network configurations (Setup 1, Setup 2, *etc.*), highlighting the efficacy and superiority of our recommended model in minimizing end-to-end latency. Reduced latency is beneficial for medical monitoring systems, as our research shows. Reducing latency allows for faster data processing, easier access to patient data, and better decision-making by healthcare professionals.

Figure 2 and Table 2 illustrate the average response time across various setups for both the cloud module and latency module using different performance configurations: PT (Setup-1), TT (Setup-2), QT (Setup-3), QoS (Setup-4), and QoE (Setup-5). The cloud module consistently shows high response efficiency, with all setups ranging from 86% to 93.5%. Notably, the QoE-based setup (Setup-5) performs the best at 93.5%, indicating an enhanced QoE due to optimised resource allocation and intelligent processing. In contrast,

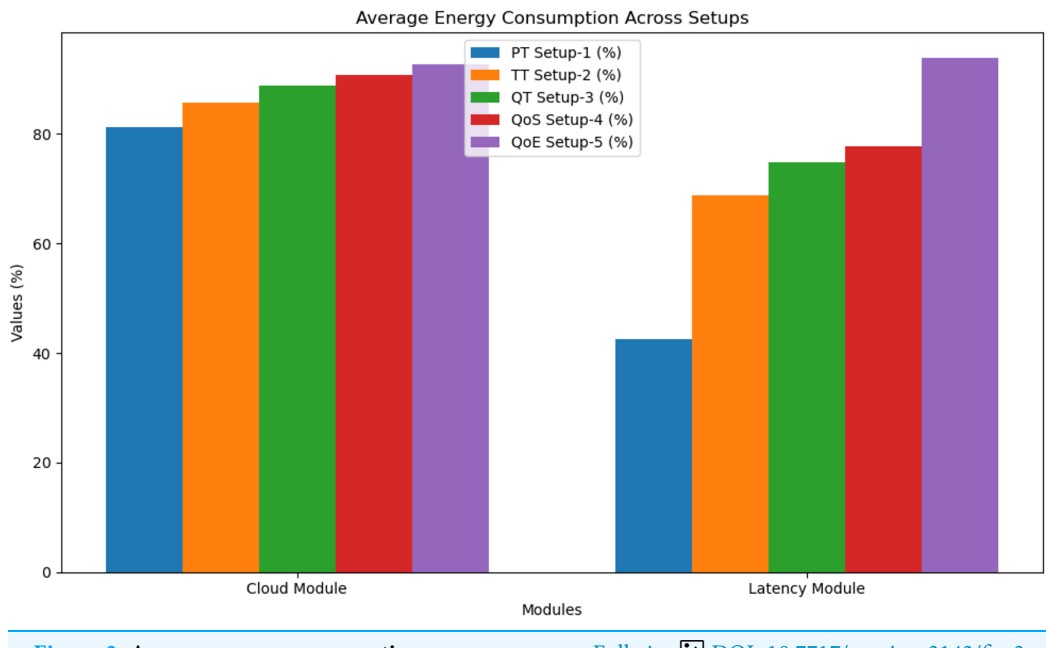

**Figure 3 Average energy consumption.**               

**Table 3 Average energy consumption.**

| Module | PT Network Setup-1 (%) | TT Network Setup-2 (%) | QT Network Setup-3 (%) | QoS Network Setup-4 (%) | QoE Network Setup-5 (%) |
|---|---|---|---|---|---|
| Cloud module | 81.2 | 85.8 | 88.9 | 90.8 | 92.8 |
| Latency module | 42.5 | 68.8 | 74.9 | 77.8 | 93.8 |

the latency module starts with a lower performance of 45% in Setup-1 but gradually improves across setups, reaching a peak of 90.5% in Setup-5. This progression demonstrates the effectiveness of incorporating QoS and QoE models to minimise latency and improve responsiveness. Overall, the result suggests that advanced configuration (QoE and QoS setups) significantly enhances performance, especially in a latency-sensitive environment, validating the benefit of applying intelligent models in fog computing architecture.

Figure 3 and Table 3 are a comparative analysis of average energy consumption across different setups: PT (Setup-1), TT (Setup-2), QT (Setup-3), QoS (Setup-4), and QoE (Setup-5) for both the cloud module and latency module.

In the cloud module, energy consumption starts at 81.2% in Setup-1 and increases gradually to a peak of 92.8% in Setup-5, indicating a trade-off between advanced optimisation and energy usage. Similarly, the latency module shows a more dramatic rise from 42.5% in Setup-1 to 93.8% in Setup-5, demonstrating that while QoE optimisation significantly enhances performance, it also results in increased energy use.

Figure 3 visually confirms these trends, with the latency module displaying a steeper energy consumption curve compared to the cloud module. Notably, the QoE-based setup
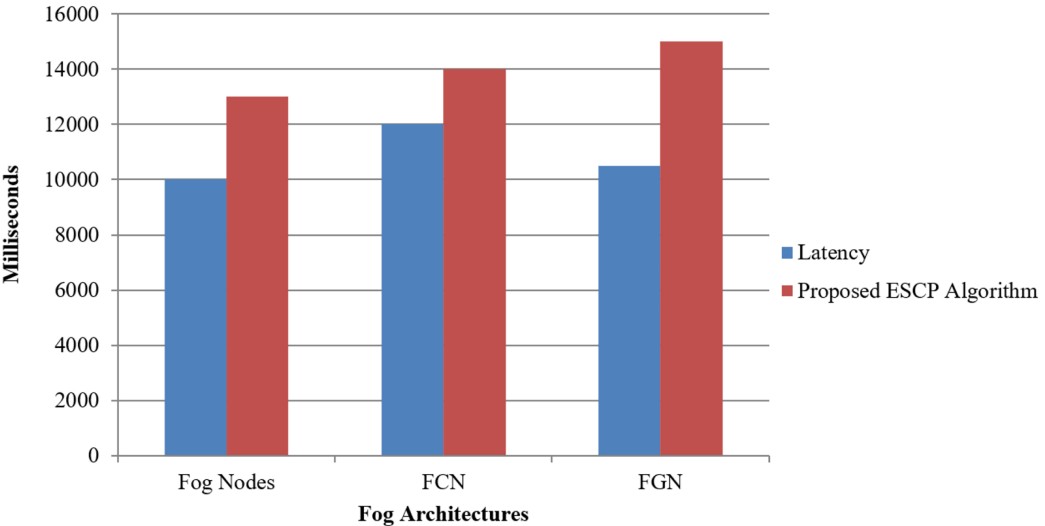

**Figure 4 Energy consumption for proposed model.**

**Table 4 Energy consumption for proposed model.**

| Gateways | Latency (ms) | Proposed ESCP algorithm (ms) |
|---|---|---|
| Fog nodes | 10,000 | 13,000 |
| FCN | 12,000 | 14,000 |
| FGN | 10,500 | 15,000 |

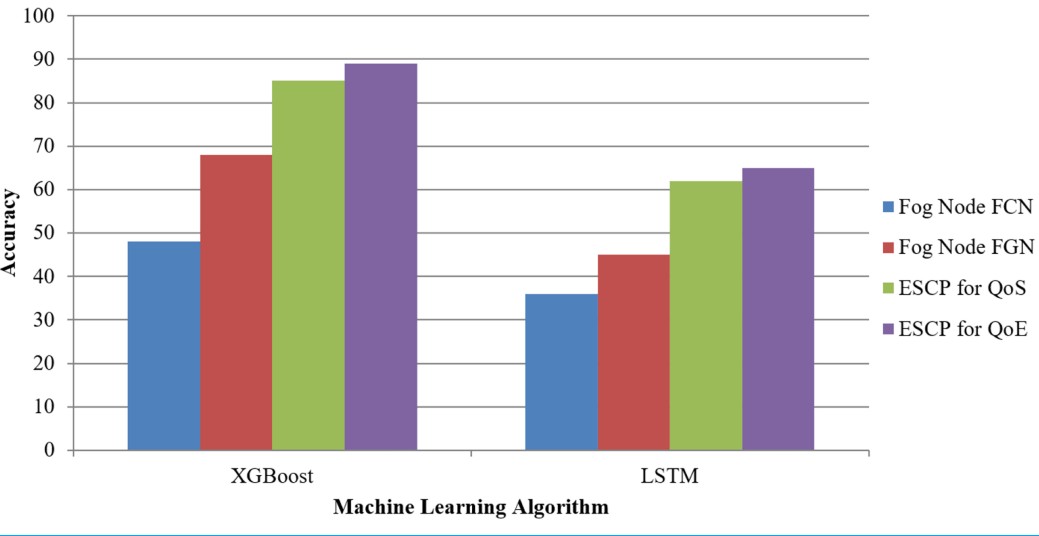

**Figure 5 Machine learning models.**

(Setup-5) consumes the most energy across both modules, but it also correlates with better responsiveness and user experience.

In summary, while QoE optimised deployments enhance overall system performance, they require more energy, especially in latency-sensitive scenarios. This insight highlights

**Table 5 Machine learning and ESCP models for FCN and FGN.**

| Algorithm | Accuracy for fog node FCN | Accuracy for fog node FGN | Accuracy for ESCP for QoS | Accuracy for ESCP for QoE |
|---|---|---|---|---|
| XGBoost | 48 | 68 | 85 | 89 |
| LSTM | 36 | 45 | 62 | 65 |

the importance of balancing energy efficiency with performance when designing intelligent fog and edge computing architectures.

Our proposed solution significantly reduces average energy consumption by distributing computing tasks across the cluster of FNs, FCN, and FGN for ESCP inside the fog layer. This energy-efficient strategy enhances the overall sustainability of fog computing systems shown in Fig. 4 and Table 4.

Figure 5, assuming it is a visual depiction of the data from the table, probably depicts how XGB and LSTM compare in these four metrics: FN-FCN, FN-FGN, ESCP for QoS, and ESCP for QoE. Fog Gateway and Computational Nodes favour XGB over LSTM for optimizing QoS and QoE metrics, as shown in Table 5. XGB beats LSTM in all circumstances.

# ANALYSIS OF RESULTS

The experimental evaluation conducted in this study compared various computing paradigms—serverless, cloud, fog, and edge through the lens of performance metrics, including latency, response time, network bandwidth, energy consumption, and failure rate. Two powerful ML models, XGB and LSTM, were employed for predicting system performance and dynamically managing resource allocation.

## Advantage

- Serverless computing demonstrates the best overall performance, particularly in terms of dynamic scalability and energy efficiency, making it highly suitable for bursts and event-driven IoT workloads.
- The proposed fog-layer system, which consists of two intelligent modules (for QoS/QoE evaluation and resource broking), showed significant improvements in latency and energy usage, outperforming traditional fog and cloud models.
- XGB offered faster and more interpretable decision-making for QoS evaluation, while LSTM effectively forecasted resource bottlenecks, allowing the system to take preemptive offloading decisions.
- The integration of AutoML led to superior model tuning, yielding better prediction accuracy than manual hyperparameter configurations and other baseline ML models.

## Disadvantages

- Although serverless computing yielded the best results, it relies heavily on stateless functions and lacks fine-grained control over hardware-level optimisation, which may be a limitation for certain real-time tasks.

- Fog and edge layers, while effective in reducing latency, face resource constraints such as limited memory and compute power, which can affect their ability to handle large-scale or compute-intensive applications.
- The LSTM model, while accurate, incurs higher computational overhead during training and interference compared to simpler models, making it less ideal for lightweight environments.

### Constraints

- The evaluation was based on simulated and controlled test beds, which may not fully reflect the unpredictable behaviour of real-world IoT deployments.
- Network variability and heterogeneous hardware diversity in real deployments could introduce deviations from the observed performance patterns.
- The system has been tested primarily for healthcare monitoring scenarios. While results are promising, further domain-specific adaptations may be required for other sectors such as industrial IoT or smart cities.

## CONCLUSION

In conclusion, this study presents a robust and scalable model that integrates ML, specifically XGB and LSTM, within serverless, fog, edge, and cloud computing environments to enhance IoT application performance. The primary contribution includes the development of a dynamic resource optimization framework using ML, a comparative evaluation of multiple computing paradigms, and the application of the model. Experimental results demonstrate that the proposed fog-layer system, composed of two modular clusters, significantly reduces latency and energy consumption compared to traditional and standalone fog/cloud systems. Additionally, the use of AutoML yielded superior performance over other ML methods in terms of accuracy and adaptability. The architecture proves especially efficient in addressing challenges such as dynamic workload scalability, battery limitations, and real-time responsiveness. For future work, we plan to extend this model to include federated learning capabilities and integrate blockchain for secure data exchange, further strengthening the model's applicability in critical IoT domains like smart healthcare and autonomous systems.

### Funding
The authors received no funding for this work.

### Competing Interests
The authors declare that they have no competing interests.

### Author Contributions
- P. Jenifer conceived and designed the experiments, performed the experiments, analyzed the data, performed the computation work, prepared figures and/or tables, authored or reviewed drafts of the article, and approved the final draft.

- J. Angela Jennifa Sujana conceived and designed the experiments, performed the experiments, prepared figures and/or tables, authored or reviewed drafts of the article, and approved the final draft.

## Data Availability

Raw data is available at Mendeley:

P, Jenifer (2025), "Dataset for QoE-aware application deployment in fog computing environments using machine learning", Mendeley Data, V1, doi: 10.17632/jnr5rc259f.1.

The QWS Dataset is GitHub: available at https://qwsdata.github.io/.

## Supplemental Information

Supplemental information for this article can be found online at http://dx.doi.org/10.7717/peerj-cs.3143#supplemental-information.

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
