# Peer review of "Quality of experience-aware application deployment in fog computing environments using machine learning"

_PeerJ Computer Science, doi:10.7717/peerj-cs.3143_

## Round 0.1 · original submission · Major Revisions

**Language Note:** The review process has identified that the English language must be improved. PeerJ can provide language editing services - please contact us at [email protected] for pricing (be sure to provide your manuscript number and title). Alternatively, you should make your own arrangements to improve the language quality and provide details in your response letter. – PeerJ Staff

Reviewer 1 ·

Basic reporting

.

Experimental design

.

Validity of the findings

.

Additional comments

Dear authors,

I reviewed the manuscript titled: “QoE-aware application deployment in fog computing environments using machine learning”, in this article, authors present an architecture that guarantees Qality of Service (QoS) and Qality of Experience (QoE). The suggested approach may use serverless, cloud, fog, and edge computing models to assess IoT-based healthcare applications. Machine learning and meta-heuristic optimization were used to investigate energy consumption, latency, bandwidth, response time, and scalability. I think that this article is interesting and the topic is relevant, however I have some concerns about the proposal.

Please check the english grammar and english details in the text.

In Fog Gateway Nodes (FGNs), authors present equation 1, I recommend describing the value range for each variable of the equation.

The section IoT Device and Edge Network is very short, I recommend including this section in another section or describe more details in this section.

In the section Machine Learning Models, authors explain a lot of details about models of the state of the art, I recommended to explain specific details about the implementation in the proposal or details that

In Figure 4, please add labels in the X and Y axis in the plot.

In Figure 5, please add labels in the X and Y axis in the plot.

In Table 3, please add units for values in the table.

Please add an analysis of the results section to explain in more details, advantages, disadvantages, best results and constraints of the results.

Please add a Figure to show more graphically the proposed architecture (devices, layers, etc.)

In conclusion, please add details about the main contributions and results obtained in the experiments, and future work.

Reviewer 2 ·

Basic reporting

The manuscript presents interesting ideas. However, the quality of the English language significantly affects the clarity and readability of the paper. I recommend that the authors seek assistance from a native English speaker or a professional editing service to improve the sentence structure and overall flow.

The abstract needs more detail regarding the importance of new AI edge devices and the necessity for low-resource requirements. It also mentions an approach proposed for healthcare applications, but the methodology and dataset sections do not demonstrate or illustrate any such usage.

Figure 1 requires more explanation, and the components "IoT Device and Edge Network" should not be combined into a single component.
Figures 2 and 3 depict setups that are not clearly explained in the paper.

The related work section lacks a thorough analysis to highlight the research gap. For example, paper [21] (line 126) is merely described without any critical analysis of the work.

The terms MEC, Edge, and IoT should be used consistently and appropriately throughout the paper.

Experimental design

There was no clear explanation of how the proposed work was designed. Figure 1 shows the proposed system architecture, but it was not well explained. Including relevant algorithms would better present the proposed work.

Validity of the findings

Results from line 307 was not well discussed.

Additional comments

The proposed system architecture was not clearly discussed, making it difficult to understand the paper's contribution. Including a written algorithm would improve clarity.

The concepts of IoT and edge networks should be clearly understood by the authors and used consistently to avoid confusion for the reader.

The results should be discussed in greater detail.

---

## Round 0.2 · accepted · Accept

I have checked the response letter and the revised manuscript. The paper can be accepted.

Reviewer 1 ·

Basic reporting

After reviewing the new version of the article, I saw that all my concerns were answered.

Experimental design

no comment

Validity of the findings

no comment

Additional comments

In my opinion, the new version of the article is clearer and the wording has been improved.